# IL-13 deficiency exacerbates lung damage and impairs epithelial-derived type 2 molecules during nematode infection

Alistair L Chenery[1,2,*] , Silvia Rosini[1,2,*] , James E Parkinson[1,2] , Jesuthas Ajendra[1,2] , Jeremy A Herrera[1] , Craig Lawless[1] , Brian HK Chan[1,2] , P'ng Loke[3] , Andrew S MacDonald[2] , Karl E Kadler[1,2] , Tara E Sutherland[2,†] , Judith E Allen[1,2,†]

**IL-13 is implicated in effective repair after acute lung injury and the pathogenesis of chronic diseases such as allergic asthma. Both these processes involve matrix remodelling, but understanding the specific contribution of IL-13 has been challenging because IL-13 shares receptors and signalling pathways with IL-4. Here, we used *Nippostrongylus brasiliensis* infection as a model of acute lung damage comparing responses between WT and IL-13-deficient mice, in which IL-4 signalling is intact. We found that IL-13 played a critical role in limiting tissue injury and haemorrhaging in the lung, and through proteomic and transcriptomic profiling, identified IL-13-dependent changes in matrix and associated regulators. We further showed a requirement for IL-13 in the induction of epithelial-derived type 2 effector molecules such as RELM-α and surfactant protein D. Pathway analyses predicted that IL-13 induced cellular stress responses and regulated lung epithelial cell differentiation by suppression of Foxa2 pathways. Thus, in the context of acute lung damage, IL-13 has tissue-protective functions and regulates epithelial cell responses during type 2 immunity.**

## Introduction

IL-13 is a central effector cytokine with diverse roles during both protective and pathogenic type 2 immune responses. During anti-parasitic immunity, IL-13 is critical for goblet cell hyperplasia and mucus production at mucosal sites (Finkelman et al, 2004). These responses are particularly essential for the expulsion of gastro-intestinal worms from the host (Shimokawa et al, 2017). However, the same mucus hypersecretion response is a hallmark of pathogenicity in asthmatic patients (Evans et al, 2009). Similarly, IL-13

can induce cytoprotective cytokines such as vascular endothelial growth factor to protect from acute lung injury (Corne et al, 2000), yet drive airway smooth muscle cell contraction leading to broncho-constrictive effects during asthma pathogenesis (Risse et al, 2011). However, IL-13 and IL-4 have overlapping signalling pathways and both use IL-4Rα. IL-4 signals through the type I receptor (IL-4Rα paired with the common γ chain) and the type II receptor (IL-4Rα paired with IL-13Rα1) whereas IL-13 only signals through the type II receptor. However, IL-13 can also ligate IL-13Rα2, which serves primarily as a non-signalling decoy receptor but may have signalling functions distinct from IL-13Rα1 (Gieseck et al, 2018; Karmele et al, 2019). Consequently, disentangling the relative roles of each cytokine in specific cell types has been challenging (Wills-Karp & Finkelman, 2008). Whereas IL-13 has been a therapeutic target for asthma with ongoing clinical trials (Bagnasco et al, 2016), dupilumab, anti-IL-4Rα which inhibits both IL-4 and IL-13 signalling, has been a front-runner treatment showing efficacy in severe asthma patients (Castro et al, 2018). Thus, understanding the individual roles of these two cytokines has important clinical implications.

Collagen deposition after tissue injury is an important aspect of wound healing and repair. However, in asthma and other chronic inflammatory conditions, dysregulated ECM remodelling and fibrosis leads to many pathological features of disease, with increased deposition of collagen and basement membrane thickening leading to a significant decline in airway function (Elias et al, 1999; Wynn, 2008). In such disease settings, a pro-fibrotic role for IL-13 is evidenced by its ability to activate pulmonary fibroblasts and stimulate fibrillar collagen synthesis (Doucet et al, 1998; Lee et al, 2001; Kolodsick et al, 2004; Chung et al, 2016). Furthermore, the role of IL-13 in promoting liver fibrosis, such as during schistosomiasis and other pathologies, is well-characterised (Kaviratne et al, 2004; Gieseck et al, 2016, 2018). However, there is an apparent context-dependent role for IL-13 in promoting pulmonary fibrosis. For instance, IL-13 is required for fibrotic changes

---

[1]Wellcome Centre for Cell-Matrix Research, Faculty of Biology, Medicine and Health, Manchester Academic Health Science Centre, University of Manchester, Manchester, UK [2]Lydia Becker Institute for Immunology and Infection, Faculty of Biology, Medicine and Health, Manchester Academic Health Science Centre, University of Manchester, Manchester, UK [3]Department of Microbiology, NYU Langone Health, New York, NY, USA

Correspondence: judi.allen@manchester.ac.uk; tara.sutherland@manchester.ac.uk
*Alistair L Chenery and Silvia Rosini contributed equally to this work
†Tara E Sutherland and Judith E Allen are joint communicating authors

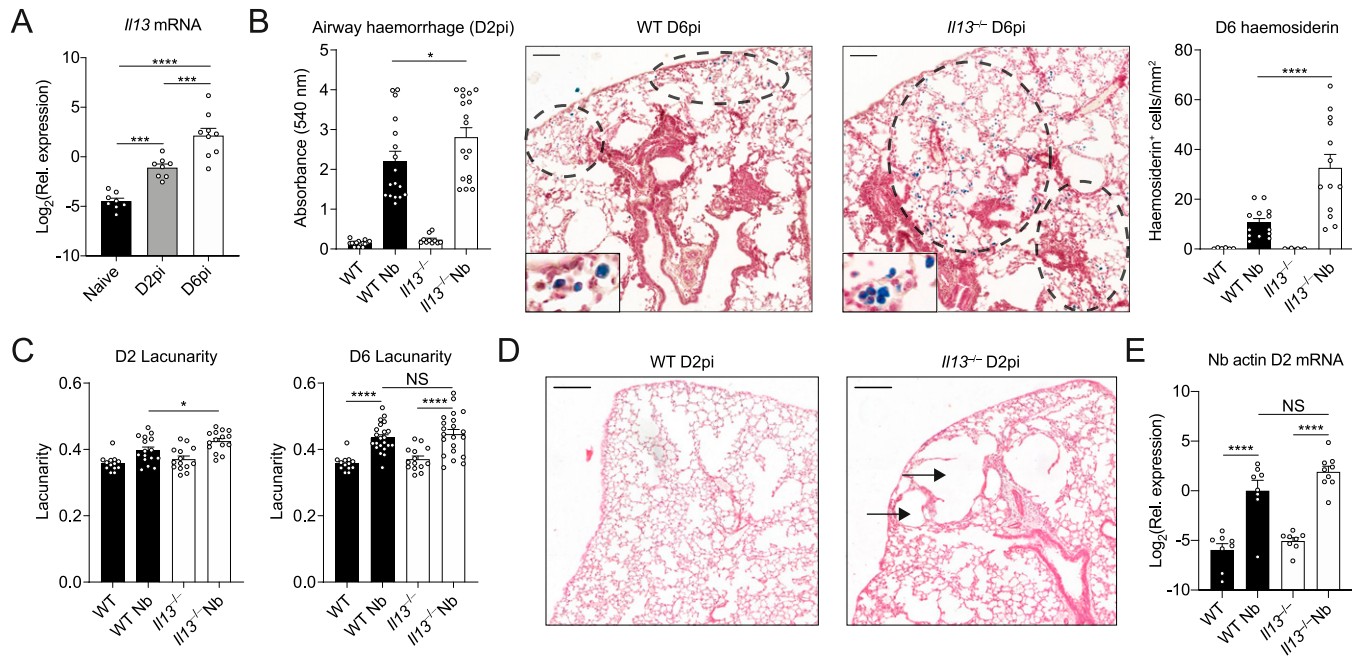

**Figure 1. Tissue-protective role for IL-13 during acute lung injury.**
WT and *Il13*$^{-/-}$ mice were infected with *Nippostrongylus brasiliensis* (Nb). **(A)** On day 2 post-infection (D2pi) and D6pi, *Il13* mRNA levels were measured in the lungs of WT mice by quantitative real-time PCR (data normalised against housekeeping gene *Rpl13a*). **(B)** On D2pi, BAL fluid absorbance at 540 nm was quantified to measure airway haemorrhage. On D6pi, lung lobe sections were stained with Prussian blue and haemosiderin-laden cells (blue) were enumerated per area of tissue (scale bar = 100 μm). **(C)** To measure airway damage on D2pi and D6pi, lacunarity for whole lung lobes was computed. **(D)** Representative haematoxylin and eosin images of infected WT and *Il13*$^{-/-}$ lungs showing alveolar damage in the tissue (scale bar = 200 μm). **(E)** Nb-specific actin mRNA in lung tissue was measured on D2pi by quantitative real-time PCR (data normalised against housekeeping gene *Rpl13a*). Data (mean ± SEM) were pooled from three individual experiments with three to six mice per group (per experiment). NS not significant, *P < 0.05, ****P < 0.0001 (one-way ANOVA and Tukey–Kramer post hoc test).

in the lung after *Schistosoma mansoni* egg challenge but not in the bleomycin model of pulmonary fibrosis (Wilson et al, 2010). Although these studies implicate IL-13 in regulating ECM components such as fibrous collagens during a variety of chronic inflammatory conditions, less is known about how IL-13 may affect the ECM and associated regulators as a whole and in acute contexts of lung injury.

In this study, we examined the function of IL-13 during the early stages of infection with the lung-migrating nematode parasite *Nippostrongylus brasiliensis*. We found that IL-13 was required for the full induction of airway eosinophilia and for limiting lung injury, independent from the other type 2 cytokines IL-4 and IL-5. IL-13 did not have a major effect on collagen dynamics during the early phase of infection. However, IL-13 was critical for the up-regulation of type 2 effector molecules involved in collagen regulation and tissue repair, such as epithelial-derived resistin-like molecule *α* (RELM-*α*). Through both proteomic and transcriptomic approaches, we provide new insight into the contribution of IL-13 to pulmonary helminth infections, in particular suggesting a broader role for IL-13 in overall type 2 immunity during acute lung injury.

# Results

### Lung injury and vascular damage is exacerbated in the absence of IL-13

Upon infection in the skin with the nematode parasite *N. brasiliensis*, larvae migrate into the circulation and by day 2 post-infection (D2pi)

burst through the lung capillaries and the airways causing extensive tissue damage and haemorrhaging (Reece et al, 2006). After infection with *N. brasiliensis*, WT mice had increased *Il13* mRNA expression in the lung on D2pi that further increased by D6pi (Fig 1A). By D6pi, T-cell activation (anti-CD3/CD28) in total lung suspensions significantly increased IL-13 protein production (Fig S1). To evaluate the role of IL-13, we used *Il13*$^{tm3.1Anjm}$ (IL-13 eGFP knock-in) mice, which are deficient for IL-13 when bred as homozygotes (henceforth referred to as *Il13*$^{-/-}$). After infection of *Il13*$^{-/-}$, we assessed acute bleeding as well the cumulative clearance of blood (Fig 1B). We first measured the bronchoalveolar lavage (BAL) fluid absorbance at 540 nm, which correlates with the increased presence of haemoglobin because of bleeding (Meng & Alayash, 2017). In the absence of IL-13, infected mice had an increased level of airway haemorrhage on D2pi relative to WT mice (Fig 1B). Efferocytosis of red blood cells has been shown to occur in the *N. brasiliensis* model (Marsland et al, 2008). Therefore, as an additional measure of bleeding, we assessed the accumulated up-take of red blood cells over time by measurement of haemosiderin within airway macrophages. Consistent with exaggerated bleeding, infected *Il13*$^{-/-}$ mice had increased numbers of haemosiderin-laden macrophages in the lung compared with controls, as determined by Prussian blue staining on D6pi (Fig 1B). To assess airway damage after infection, we evaluated H&E–stained lung sections with lacunarity analysis to determine gaps in alveolar structures (Chenery et al, 2019). By D2pi, *Il13*$^{-/-}$ mice had increased lacunarity scores relative to WT (Fig 1C). Upon gross inspection of lung sections from D2pi, it was evident that infected *Il13*$^{-/-}$ lungs had larger gaps in the alveoli, presumably in

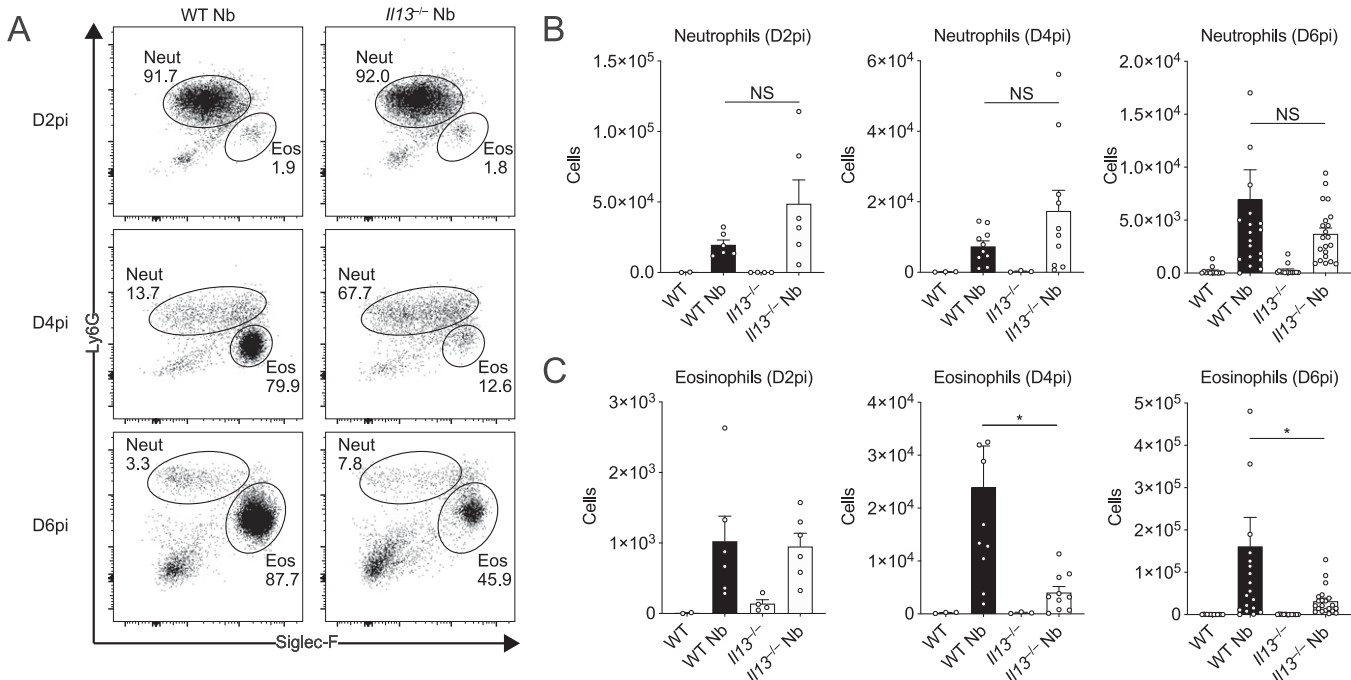

**Figure 2. IL-13-dependent airway eosinophilia during *Nippostrongylus brasiliensis* infection.**
WT and *Il13*⁻/⁻ mice were infected with *N. brasiliensis* (Nb) and BAL cells were analysed by flow cytometry. **(A)** Representative plots of percentages of BAL CD11c⁻CD11b⁺Ly6G⁺ neutrophils and CD11c⁻CD11b⁺Siglec-F⁺ eosinophils on D2, D4, and D6pi. Numbers indicate percentage of cells within total live CD45.2+ cells. **(B, C)** Total BAL neutrophil and (C) eosinophil cell counts on D2, D4, and D6pi. Data (mean ± SEM) were representative (day 2 post-infection) or pooled (D4 and D6pi) from four individual experiments with three to five mice per group (per experiment). NS: not significant, *P < 0.05 (one-way ANOVA and Tukey–Kramer post hoc test).

areas proximal to where larvae burst through the tissue (Fig 1D). However, by D6pi lacunarity was comparable between *Il13*⁻/⁻ and WT mice (Fig 1C). In primary infection, IL-13 is only involved in parasite expulsion after the parasites have already cleared the lung tissue (Bouchery et al, 2015). Nonetheless, we assessed the possibility that the exacerbated damage on D2pi in *Il13*⁻/⁻ mice was due to an increased number of larvae entering the lungs in the absence of IL-13. Analysis of *Nippostrongylus*-specific actin mRNA levels revealed comparable lung-stage parasites between infected WT and mice on D2pi (Fig 1E) These data strongly suggest an early host tissue-protective role for IL-13 in partially limiting airway haemorrhaging and tissue damage immediately after *N. brasiliensis* entry into the lung.

## IL-13 is required for airway eosinophilia during *N. brasiliensis* infection

Neutrophilia is a major contributor to acute lung injury and hae-morrhage during helminth infection (Chen et al, 2012). At D2pi, neutrophils are the predominant infiltrating granulocyte and con-tribute to worm killing but at the expense of host tissue injury (Chen et al, 2012; Sutherland et al, 2014). We hypothesised that the exacerbated bleeding and damage in *Il13*⁻/⁻ mice was due to the requirement for IL-13 to suppress the early neutrophilia. Thus, we aimed to characterise the role of IL-13 during the early granulocyte response in the lungs. Using *N. brasiliensis* infection of *Il13*⁻/⁻ mice, we found that the proportion of infiltrating neutrophils in the BAL was not significantly altered in the absence of IL-13 when compared

with infected WT mice on D2pi (Fig 2A). In terms of absolute numbers, *Il13*⁻/⁻ mice only exhibited a slight trend for increased neutrophils compared with controls on D2pi and D4pi (Fig 2B). Between D4-6pi, the parasite larvae have exited the lungs, en route to the small intestine and the granulocyte response shifts towards eosinophilia in the airways. As early as D4pi, *Il13*⁻/⁻ mice had a major reduction in the proportion and number of eosinophils in the BAL relative to infected WT mice and this response was sustained at D6pi (Fig 2A and C). This major reduction in eosinophil proportions in infected *Il13*⁻/⁻ mice accounted for the apparent increase in neutrophil percentage at D4pi and D6pi. Together, these results show that IL-13 is required for the full induction of airway eosin-ophilia after *N. brasiliensis* infection.

## IL-4 and IL-5 do not compensate in the absence of IL-13 after *N. brasiliensis* infection

Previous studies using the *N. brasiliensis* model in IL-4Rα–deficient animals or IL-4/IL-13-double–deficient mice did not distinguish between the relative roles of IL-4 and IL-13 during infection (Urban et al, 1998; Mearns et al, 2008; Oeser et al, 2015). Unlike those settings, in IL-13 cytokine–deficient mice which have intact type I and II IL-4 receptors, IL-4 may compensate for IL-13 deficiency. To evaluate the potential role of IL-4, we first assessed overall sus-ceptibility to *N. brasiliensis* in the small intestine which is known to be dependent on both IL-4Rα and IL-13Rα1 (i.e., type I and II IL-4 receptors) (Barner et al, 1998; Urban et al, 1998). Where WT mice had largely cleared their parasites, all *Il13*⁻/⁻ mice harboured adult

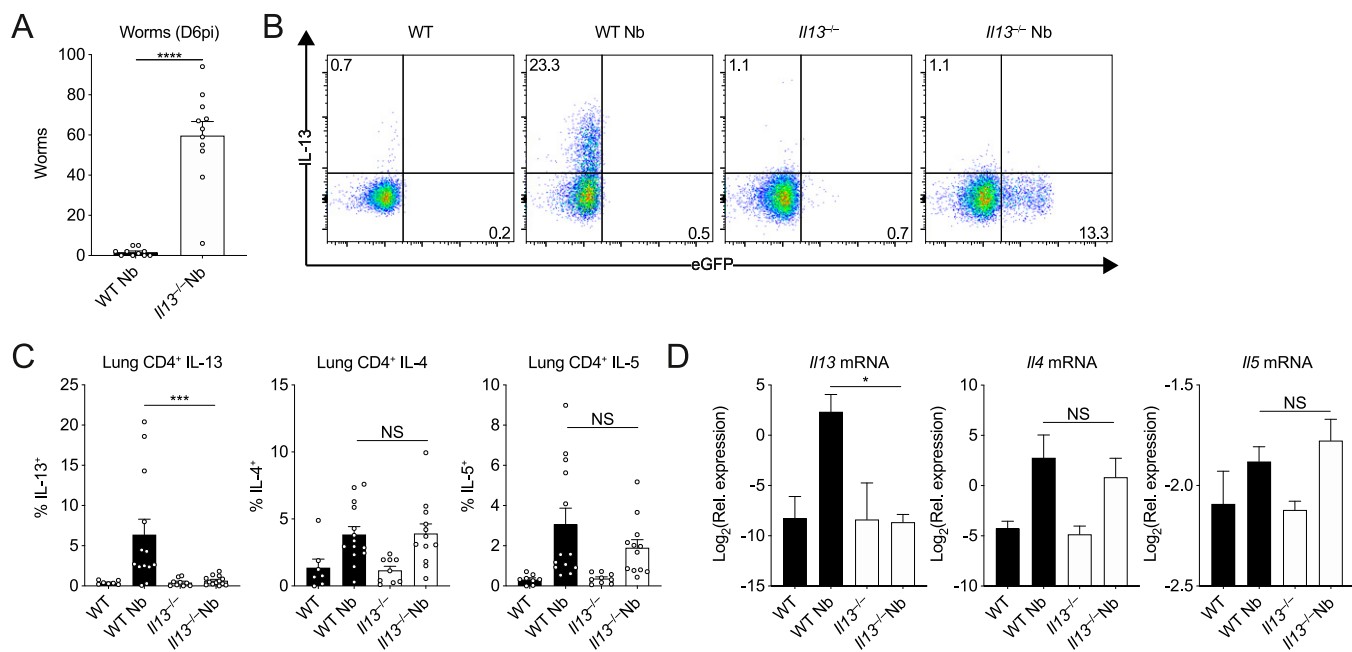

**Figure 3. IL-4 and IL-5 do not compensate in the absence of IL-13 during *Nippostrongylus brasiliensis* infection.**
WT and *Il13*⁻/⁻ mice were infected with *N. brasiliensis* (Nb). **(A)** On D6pi, adult worms in the small intestine were quantified. **(B)** Representative flow cytometry plots of lung CD4⁺ T cells stimulated ex vivo to measure intracellular WT IL-13 and KO eGFP expression. **(C)** Percentages of CD4⁺ T cells expressing IL-13, IL-4, and IL-5. **(D)** Whole lung *Il13*, *Il4*, and *Il5* mRNA was measured by quantitative real-time PCR (data normalised against housekeeping gene *Rpl13a*). Data (mean ± SEM) were pooled from three individual experiments with three to five mice per group (per experiment). NS: not significant, *P < 0.05, ***P < 0.001, ****P < 0.0001 (one-way ANOVA and Tukey–Kramer post hoc test).

intestinal worms on D6pi (Fig 3A). However, IL-13 can play a role in the differentiation of Th2 cells (McKenzie et al, 1998b), and it was therefore possible that increased worm burden in IL-13 deficient mice was due to impaired Th2-cell activation after infection. We therefore assessed lung CD4⁺ T cells on D6pi by ex vivo stimulation and measurement of intracellular type 2 cytokine expression. We confirmed that lung CD4⁺ T cells in our *Il13*⁻/⁻ mouse strain expressed eGFP in lieu of functional IL-13 after infection with *N. brasiliensis* when compared with controls (Fig 3B). Importantly, in the absence of IL-13 expression, neither IL-4 nor IL-5 cytokine expression was changed in CD4⁺ T cells after infection (Fig 3C). Furthermore, measurement of type 2 cytokine mRNA from whole lung also showed no change in the expression of *Il4* and *Il5* between infected *Il13*⁻/⁻ and WT mice (Fig 3D). Thus, IL-13 cytokine-deficient mice become fully susceptible to *N. brasiliensis* infection with no evidence that altered susceptibility is due to reduced IL-4/IL-5 during the adaptive type 2 response.

## Lung proteomic analysis after *N. brasiliensis* infection

Our results thus far showed that IL-13 has a role in limiting lung injury and enabling eosinophil recruitment into the airways after *N. brasiliensis* infection. Although gross changes in lung structure were not evident at day 6 between WT and *Il13*⁻/⁻ mice (Fig 1C), it is feasible that changes to physical lung injury in the absence of IL-13, as was seen on D2 post-infection (Fig 1B and C), could have profound effects on the way in which the lung repairs compared with WT animals. To directly address this possibility at the whole tissue level, we performed mass spectrometry on H&E–stained lung

sections at D6pi. In infected WT mice, 648 proteins were significantly changed (adjusted *P*-value < 0.05) relative to uninfected mice with most of these proteins being up-regulated (Fig 4A). We then specifically analysed changes in the ECM and matrix-related proteins (defined from the Matrisome Project [Naba et al, 2012]), performing hierarchical clustering across groups after *N. brasiliensis* infection (Fig 4B). Several collagen types (notably collagens I, III, and VI) clustered together, with proteins reduced in WT mice after infection, an effect not replicated in *Il13*⁻/⁻ mice. Such differences may relate to an already lower level of expression of these collagens under basal conditions in *Il13*⁻/⁻ versus WT mice. Evaluation of the relative abundance of collagen types across groups confirmed dysregulated collagen levels in the absence of IL-13, with *N. brasiliensis* infection altering collagen expression in WT but not *Il13*⁻/⁻ mice (Fig 4C). Fibrillar collagens I and III, although already low in *Il13*⁻/⁻ mice, were decreased in infected WT mice and basement membrane-associated collagens IV and VI were also dysregulated in *Il13*⁻/⁻ mice. To visualise total collagen deposition, Masson's trichrome staining was performed but did not show any major differences between infected WT and *Il13*⁻/⁻ mice at D6pi (Fig S2A). Furthermore, hydroxyproline levels were measured as a proxy for total collagen in the lung and confirmed no gross differences between the groups (Fig S2B). To gain further insight into global IL-13–dependent changes in the lung, we performed pathway analysis on differentially expressed proteins (by non-adjusted *P*-value < 0.05) comparing infected WT and *Il13*⁻/⁻ mice (Fig 4D). IL-13-dependent canonical pathway analysis showed a down-regulation (blue) in pathways relating to protein synthesis and cellular stress after infection, with 71% of endoplasmic reticulum stress pathway-associated

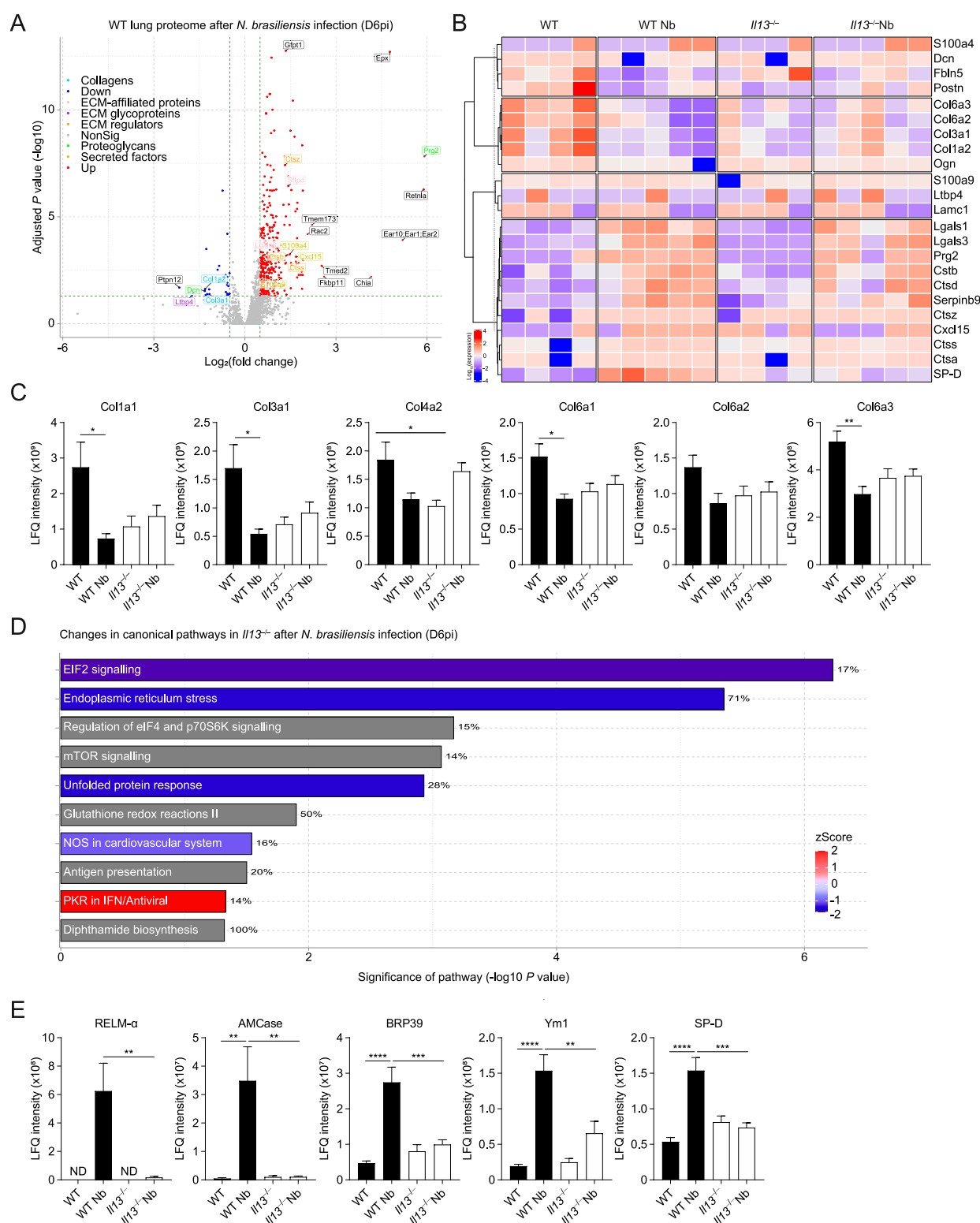

**Figure 4. IL-13–dependent lung proteomic changes during *Nippostrongylus brasiliensis* infection.**
WT and *Il13*−/− mice were infected with *N. brasiliensis* (Nb) and on D6pi lungs were prepared for proteomic analysis. **(A)** Volcano plot of infected WT lungs (D6pi) showing differential expression of up- (red) or down- (blue) regulated proteins with matrisome annotations (fold change relative to naïve WT mice). Black labels indicate proteins lacking matrisome annotations. **(B)** Unsupervised, hierarchically clustered heatmap of expression of matrisome proteins comparing naïve and Nb-infected groups of WT and *Il13*−/− mice on D6pi. **(C)** Columns in each set represent different (biological repeat) mice in each group (C) Relative abundance (label-free quantification [LFQ] intensity) of collagen peptides comparing infected WT and *Il13*−/− lungs on D6pi. **(D)** Predicted changes in canonical pathways based on changes in the proteome of infected *Il13*−/− mice compared with infected WT mice (down-regulation in blue, up-regulation in red, and no specified direction in grey). Percentage indicates the relative

proteins being regulated by IL-13. In addition, mTOR signalling and antigen presentation pathways appeared to be altered in the absence of IL-13. Although proteomic analysis of lungs from infected $Il13^{-/-}$ mice did not reveal any differences in matrisome components compared with infected WT mice, RELM-$\alpha$ and surfactant protein D (SP-D), two proteins heavily implicated in type 2 immunity (Thawer et al, 2016; Sutherland et al, 2018), were significantly decreased in infected $Il13^{-/-}$ lungs amongst the total proteome (Fig S3). Several other key proteins were found to be significantly down-regulated in $Il13^{-/-}$ mice based on relative abundance when compared with infected WT mice including AMCase, BRP39, and Ym1 (Fig 4E) molecules also strongly associated with pulmonary type 2 immunity (Sutherland et al, 2014; Kim et al, 2015). These data suggest that IL-13 may not directly regulate ECM remodelling after acute lung injury on D6pi after *N. brasiliensis* infection. However, IL-13 is critical for the induction of type 2 effector molecules which may determine the subsequent tissue repair responses.

### IL-13 is required for induction of epithelial cell–derived RELM-$\alpha$

The proteomic analysis led us to further characterise the contribution of IL-13 to RELM-$\alpha$ expression in the lung. RELM-$\alpha$ is an important type 2-associated effector molecule implicated in repair processes that have been previously described in the skin and lungs (Knipper et al, 2015; Sutherland et al, 2018). We measured RELM-$\alpha$ protein in the BAL fluid by ELISA (i.e., release into the airways) and consistent with our proteomics data found decreased RELM-$\alpha$ levels in $Il13^{-/-}$ mice infected with *N. brasiliensis* relative to infected WT mice (Fig 5A). Similarly, quantification of mRNA in the whole lung showed that the induction of *Retnla* expression on D4pi and D6pi was reduced in the absence of IL-13 (Fig 5B). In addition, we performed immunofluorescence staining in lung sections and found that the airways, which were highly RELM-$\alpha^+$ after infection in WT mice, appeared largely diminished in RELM-$\alpha$ expression in infected $Il13^{-/-}$ mice (Fig 5C). To directly quantify this cellular RELM-$\alpha$ expression, lung cell suspensions were analysed by flow cytometry for intracellular RELM-$\alpha$. In the absence of IL-13, CD45$^-$EpCAM$^+$ epithelial cells had significantly impaired expression of RELM-$\alpha$ as early as D2pi and was still muted by D6pi when compared with WT controls (Fig 5D). Consistent with previous findings (Sutherland et al, 2018; Krljanac et al, 2019), epithelial cells were the major source of RELM-$\alpha$ at these time points, whereas other cellular sources such as alveolar macrophages, neutrophils, and eosinophils, were relatively unchanged in the absence of IL-13 after infection (data not shown). As a complementary experiment, equimolar amounts of systemic IL-4 and IL-13 were each delivered by intraperitoneal injection into WT mice and 18 h later epithelial RELM-$\alpha$ expression was measured in the lungs. Both IL-4 and IL-13 injection elicited a comparable expression of RELM-$\alpha$ in tissue resident (F4/80$^{hi}$) macrophages in the peritoneal cavity (Fig S4A). In contrast to IL-4 which had no apparent effect, systemic IL-13 delivery was able to potently drive RELM-$\alpha$ expression in CD45$^-$EpCAM$^+$ epithelial cells (Fig 5E). However, we hypothesized that the

differences in response to comparable doses of IL-4 versus IL-13 may be due to bioavailability in the lung when delivered systemically. Therefore, we directly administered the cytokines to the lung via intranasal instillation and found that both IL-4 and IL-13 induced epithelial cell derived–RELM-$\alpha$ (Fig 5F). Analysis of non-epithelial cells revealed a similar pattern of RELM-$\alpha$ expression in airway macrophages in response to IL-4 versus IL-13 when comparing intraperitoneal and intranasal delivery (Fig S4B and C). Taken together, these results show that IL-13 is both necessary and sufficient to stimulate lung epithelial cell RELM-$\alpha$. Comparison of intranasal delivery with peritoneal delivery of IL-4 versus IL-13 suggest either that lung epithelial cells and airway macrophages are more sensitive to IL-13 relative to IL-4, or that IL-13 traffics more readily to the lung than IL-4, perhaps because IL-4 is consumed along the way by the more abundant type 1 IL-4 receptors.

### IL-13 broadly regulates type 2 immunity in the lung during *N. brasiliensis* infection

Because of the technical limitations of proteomics not being able to detect low molecular weight cytokines and chemokines, we performed transcriptional profiling of lung tissue from *N. brasiliensis*–infected WT and $Il13^{-/-}$ mice to better define the role of IL-13 during type 2 immunity in acute injury settings and to highlight potential mechanisms/pathways involved. Using the Nanostring Myeloid Innate Immunity v2 panel, we screened for differentially regulated genes across groups on D6pi. Principal component analysis showed distinct separation between groups based on infection status (PC1) and genotype (PC2) (Fig 6A). Genes were grouped by unsupervised hierarchical clustering and differentially expressed genes were represented as a heat map to identify expression patterns across all groups (Fig 6B). Notably, various signature type 2 genes were down regulated in infected $Il13^{-/-}$ mice such as *Chil3/4*, *Arg1*, *Il33*, *Retnla*, and the eotaxins (encoded by *Ccl11* and *Ccl24*). Conversely, a cluster of pro-inflammatory genes that included *Mmp8*, *Il18*, and *Tlr2* were up-regulated in infected IL-13–deficient mice (Fig 6B). Differentially expressed genes were further characterised using Ingenuity Pathway Analysis to identify potential upstream regulators (Fig 6C). Very few regulators were predicted to be up-regulated during IL-13 deficiency but included the airway epithelial cell-associated transcription factor Foxa2 (Wan et al, 2004) and the adenosine A1 receptor, Adora1. To validate changes in the Foxa2 pathway in the absence of IL-13 during *N. brasiliensis* infection, we analysed expression of genes known to be regulated by Foxa2 during type 2 settings in the lung (Chen et al, 2010). *Clca1*, *Muc5ac*, and *Ccl11* were confirmed to be highly up-regulated during infection in WT mice but impaired in $Il13^{-/-}$ mice (Fig 6D). Notably, baseline expression of *Clca1* and *Foxa3* were lower in the lungs of naïve $Il13^{-/-}$ mice when compared with controls. Foxa2 also negatively regulates *Il33* expression (Chen et al, 2010), and whereas there was a trend toward *Il33* down-regulation on D6pi in the absence of IL-13, this did not reach significance. To better

---

number of proteins that are regulated in each pathway. **(E)** Relative abundance of peptides highly associated with type 2 immunity comparing infected WT and $Il13^{-/-}$ lungs on D6pi. Data (mean ± SEM in C and E) are pooled from two independent mass spectrometry runs with four to five mice per group in total. *$P < 0.05$, **$P < 0.01$, ***$P < 0.001$, ****$P < 0.0001$ (one-way ANOVA and Tukey–Kramer post hoc test).

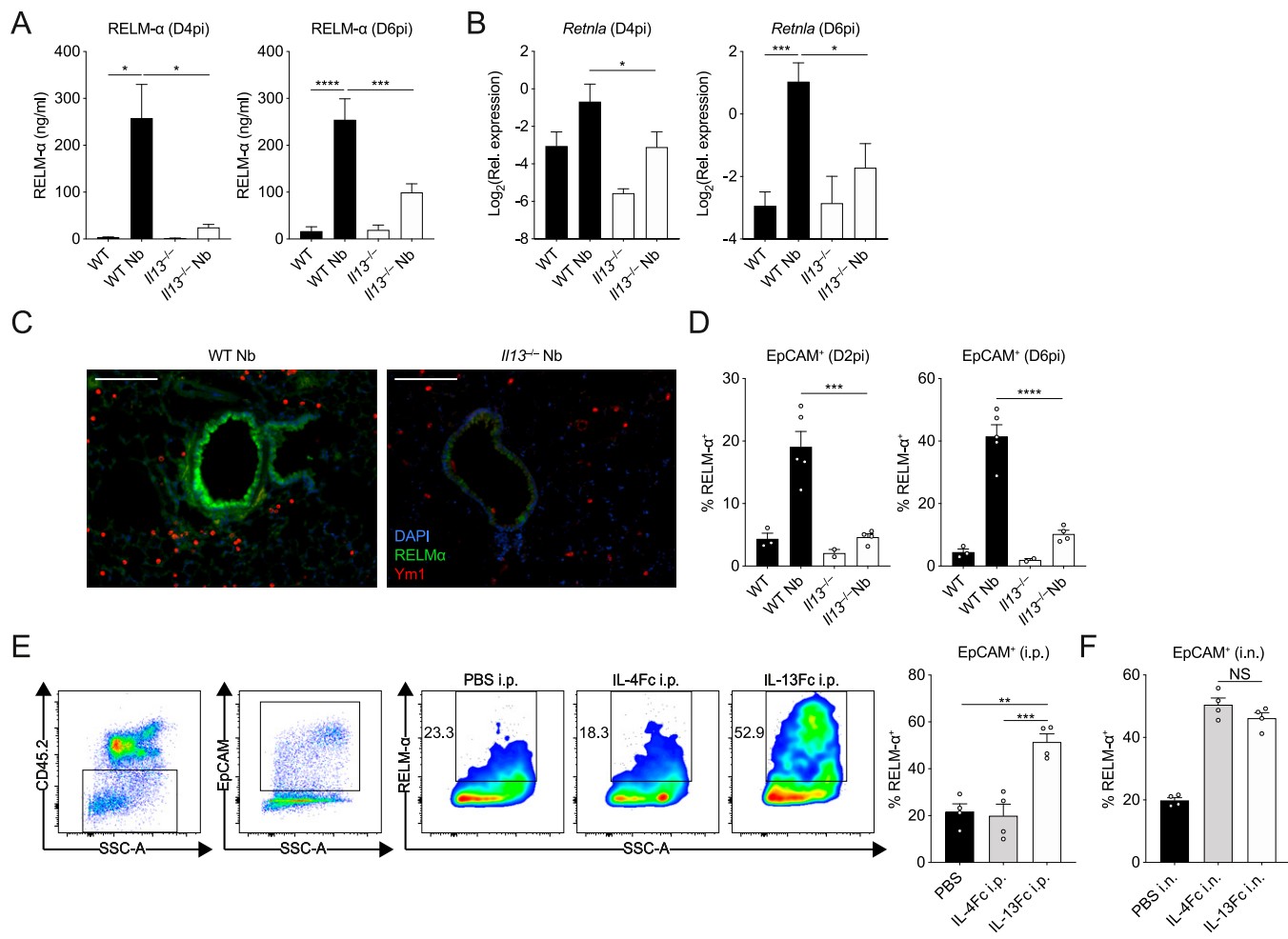

**Figure 5.  Lung epithelial cell expression and airway release of RELM-α is IL-13 dependent.**
WT and *Il13*[−/−] mice were infected with *Nippostrongylus brasiliensis* (Nb). **(A)** On D4 and D6pi RELM-α protein levels in the BAL fluid were measured by ELISA. **(B)** D4 and D6pi whole lung *Retnla* mRNA was measured by quantitative real-time PCR (data normalised against housekeeping gene *Rpl13a*). **(C)** Lung RELM-α (green) and Ym1 (red) were imaged by immunofluorescence microscopy (scale bar = 100 μm). **(D)** On D2 and D6pi, CD45[−]EpCAM[+] lung epithelial cells were analysed and quantified by flow cytometry to measure intracellular RELM-α. **(E, F)** WT mice were injected with either PBS, IL-4Fc, or IL-13Fc i.p. or (F) i.n. and 18 h later, CD45[−]EpCAM[+] lung epithelial cell RELM-α expression was measured by flow cytometry. **(A, B)** Data (mean ± SEM) in (A, B) were pooled from three individual experiments with three to five mice per group (per experiment). **(C, D, E, F)** Data (mean ± SEM) in (C, D, E, F) were representative of two individual experiments with two to five mice per group (per experiment). NS: not significant, *P < 0.05, **P < 0.01, ***P < 0.001, ****P < 0.0001 (one-way ANOVA and Tukey–Kramer post hoc test).

characterise IL-33 expression, we performed immunofluorescence staining of nuclear protein in the lungs following *N. brasiliensis* infection (Fig 6E). In WT mice expression of nuclear IL-33 was increased in the lung parenchyma, likely alveolar type II epithelial cells, on D2pi when compared with uninfected mice. In contrast, there was no significant increase in nuclear IL-33 in these cells in the absence of IL-13. In summary, these data demonstrate a broad role for IL-13 in regulating type 2 immunity and epithelial cell function during acute helminth infection in the lung.

## Discussion

Despite sharing receptors/signalling components with IL-4, various studies have established a distinct role for IL-13 during type 2

immunity and argued against simple functional redundancy between these two cytokines. For example, IL-4 cytokine–deficient mice, but not IL-4Rα-deficient mice, are able to expel *N. brasiliensis* from the gut because of intact IL-13 signalling (Barner et al, 1998). In addition, IL-13-deficient mice fail to clear *N. brasiliensis* parasites from the gut when compared with IL-4–deficient and WT mice (McKenzie et al, 1998a). Although these studies establish a clear role for IL-13 in mediating type 2 immunity in the small intestine and the establishment of the adaptive immune response, very little is known about the functions of IL-13 in the earlier lung stage of this infection model. Our study reinforces these earlier findings but also reveal a crucial protective role for IL-13 in limiting acute lung injury, promoting airway eosinophilia, and inducing type 2 effector proteins. These lung-specific effects of IL-13 are also consistent with other work showing that pulmonary eosinophilia is impaired in IL-13Rα1–deficient mice during asthma (Kumar et al, 2002; Munitz et al,

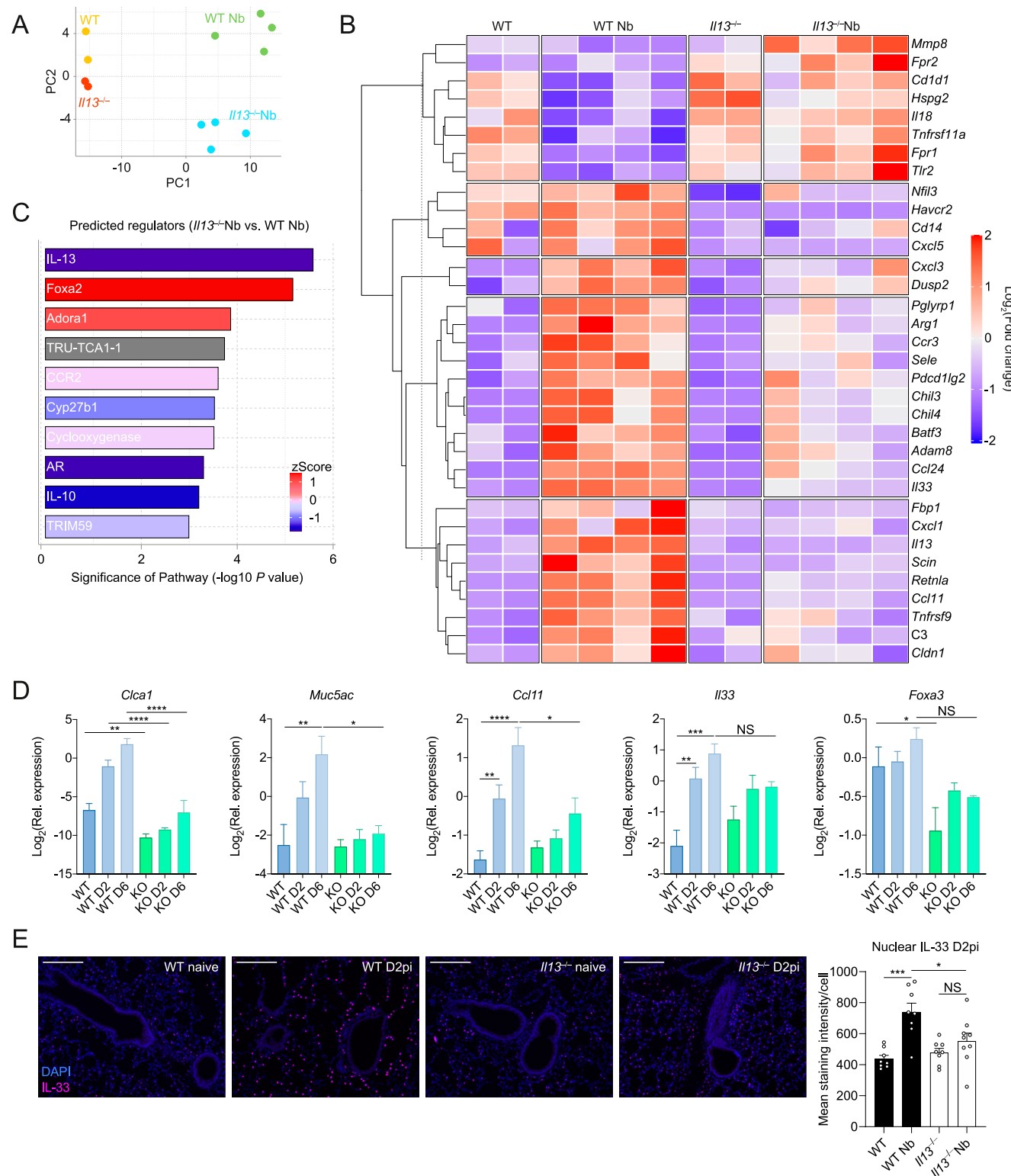

**Figure 6. Transcriptional profiling of IL-13–dependent genes in the lung after *Nippostrongylus brasiliensis* infection.**
Whole lung RNA from WT and *Il13⁻/⁻* mice infected with *N. brasiliensis* (Nb) on D6pi was analysed by Nanostring. **(A)** Principle components analysis of naïve and infected WT and *Il13⁻/⁻* mice. **(B)** Unsupervised, hierarchically clustered heat map of genes differentially expressed between mouse groups with fold change expression level indicated by colour. **(C)** Columns in each set represent different (biological repeat) mice in each group (C) Predicted upstream regulators from Ingenuity Pathway Analysis. **(D)** Expression of Foxa2-regulated genes *Clca1, Muc5ac, Ccl11, Il33,* and *Foxa3* were measured in lung tissues on day 2 post-infection and D6pi by quantitative real-time PCR (data normalised against housekeeping gene *Rpl13a*). **(E)** Immunofluorescence staining of nuclear IL-33 (magenta) in the parenchyma of the lung and quantification

2008). Our results are also highly consistent with a study by Karo-Atar et al (2016) that revealed marked epithelial-specific defects in the lungs of IL-13Rα1–deficient mice under baseline conditions and during bleomycin-induced pulmonary injury (Karo-Atar et al, 2016). These IL-13-dependent alterations included many proteins identified in our study including Clca1, RELM-α, Arginase1, MMP8, and chitinase-like proteins. This study also found that IL-13Ra1 deficiency led to increased bleomycin-induced pathology and together with our data highlights the importance of the IL-13/IL-13Rα1 axis in reducing lung injury.

Our data demonstrate a profound epithelial cell–specific effect of IL-13 during acute lung injury with *N. brasiliensis*. Expression of the type I and type II IL-4 receptors is restricted based on cell type. Hematopoietic cells predominantly express the type I receptor, whereas non-hematopoietic cells such as epithelial cells primarily express the type II receptor (Bao & Reinhardt, 2015). Whereas the type II receptor can be ligated by both IL-4 and IL-13, IL-13 can outcompete IL-4 by more efficiently promoting receptor assembly (LaPorte et al, 2008). In terms of epithelial cell expression, a model of mechanical injury shows that IL-13Rα1 expression increases at the wound edge in cultured alveolar epithelial cells (White et al, 2010). As we observed exacerbated airway injury in *Il13*$^{-/-}$ mice, it is therefore plausible in our model of nematode-induced lung injury that early alveolar epithelial cell function is dependent on the presence of IL-13. Our study also highlights impaired expression of type 2 effector molecules RELM-α and SP-D by lung epithelial cells in the absence of IL-13. RELM-α is an IL-4Rα–dependent protein involved in lung tissue repair and has been shown to mediate collagen cross-linking via lysl hydroxylase 2 (Knipper et al, 2015; Sutherland et al, 2018). Interestingly, SP-D has known roles in promoting immunity to *N. brasiliensis* and is required for the up-regulation of RELM-α in the lung (Thawer et al, 2016). Furthermore, our transcriptional profiling predicted a dysregulation of the transcription factor Foxa2 in the absence of IL-13 after infection. Foxa2 is required for alveolarization and negatively regulates goblet cell hyperplasia (Wan et al, 2004). In terms of type 2 function, the predicted increased Foxa2 activity in *Il13*$^{-/-}$ mice is consistent with previous studies showing that IL-13 decreases Foxa2 expression to enable mucin production by airway epithelial cells (Zhen et al, 2007; Park et al, 2009; Chen et al, 2010). This is also consistent with our finding that IL-33 expression was impaired in the absence of IL-13, likely via Foxa2 as has been previously shown (Chen et al, 2010). We therefore hypothesize a mechanism of IL-13–mediated suppression of Foxa2 in alveolar epithelial cells that enables the expression and release of type 2 effector molecules such as RELM-α. It is also notable that MMP-8 and IL-18 were up-regulated in infected *Il13*$^{-/-}$ mice. MMP-8 and IL-18 are pro-inflammatory markers associated with chronic obstructive pulmonary disease (COPD) (Ilumets et al, 2007; Imaoka et al, 2008). As *N. brasiliensis* infection eventually results in emphysema (Marsland et al, 2008) that resemble features of COPD, our data thus suggest a complex role for IL-13 in the development of emphysema.

Given the attribution of IL-13 to tissue remodelling and fibrosis in a variety of contexts, we anticipated changes to the matrisome in our acute lung injury model using proteomic analysis. Although we saw many changes to the lung matrisome due to infection in WT mice, we did not observe major changes in lung collagens in *Il13*$^{-/-}$ mice after infection. However, we saw that some major collagens (especially collagen I and III) were reduced in abundance after infection of WT mice. This reduction in the expression of these collagens did not occur in *Il13*$^{-/-}$ mice, which already had a baseline decrease in these collagens. It is thus possible that IL-13 has a developmental role in collagen organisation that predisposed the *Il13*$^{-/-}$ mice to enhanced tissue injury. A limitation of our matrisome analysis is that we looked at only a single time point (D6pi) and did not account for potential changes in the localisation of specific collagen types and cannot exclude a role for glycosylation state or other post-translational modifications. Nonetheless, pathway analyses of the proteomic data predicted a dysregulation in protein synthesis and cellular stress pathways (presumably in epithelial cells) upon acute tissue injury in the absence of IL-13, which will be the subject of future studies.

Our observation that *Il13*$^{-/-}$ mice had enhanced haemorrhaging led us to hypothesize that there may be compromised endothelial cell integrity (e.g., with respect to basement membrane collagen composition) during lung injury in the absence of IL-13. It is worth noting that Chen et al (2012) showed increased bleeding in infected IL-4Rα–deficient mice but not IL-13Rα1 KO suggesting IL-4 alone is sufficient to limit bleeding. This difference with our study is likely due background strain of mice used; BALB/c in the Chen et al (2012) study versus C57BL/6 mice in our study. In addition to well-known differences in IL-4 and IL-13 levels and responsiveness between strains, we routinely observe that BALB/c are much more prone to bleeding, consistent with reports of enhanced pulmonary hae-morrhage in BALB/c versus C57BL/6 mice (Fisher et al, 2016). Although we have yet to unravel these differences mechanistically, IL-4Rα signals are important in vascular integrity (Knipper et al, 2015), which could explain differential requirements for IL-4 versus IL-13 to limit bleeding between the two strains. In conclusion, our study has demonstrated a pivotal role for IL-13 in limiting tissue injury and airway bleeding and suggests broader functions for IL-13 in regulating type 2 immunity, in the context of acute lung damage.

# Materials and Methods

### Mice and ethics statements

WT C57BL/6J were purchased from Charles River UK. *Il13*$^{tm3.1Anjm}$ (Neill et al, 2010) were maintained on a C57BL/6J background and bred in-house at the University of Manchester. Most experiments had a combination of purchased WT mice and littermate controls. Female and male mice were used at age 8–14 wk. Animals were housed in individually ventilated cages with food and water

---

of mean integrated density (IntDen) (scale bar = 100 μm). Data in (A, B, C) are from a single Nanostring run with samples from two to four mice per group. Data (mean ± SEM) in (D, E) were pooled from two individual experiments with three to five mice per group (per experiment). NS, not significant, *$P < 0.05$, **$P < 0.01$, ***$P < 0.001$, ****$P < 0.0001$ (one-way ANOVA and Tukey–Kramer post hoc test).

**Life Science Alliance**

provided ad libitum. Experimental mice were randomly assigned to groups. All experiments were carried out in accordance with the United Kingdom Animals (Scientific Procedures) Act 1986 and under a Project License (70/8548) granted by the Home Office and approved by local Animal Ethics Review Group at the University of Manchester.

### N. brasiliensis infection

N. brasiliensis worms were propagated as previously described (Lawrence et al, 1996). Infective L3 larvae were isolated and mice were injected with 250 L3s subcutaneously. Mice were culled by overdose of pentobarbitone i.p. and BAL was performed with 10% FBS in PBS and lung lobes were collected. Perfusion was not performed because of the compromised lung vasculature during N. brasiliensis infection. On D2pi, BAL fluid absorbance was measured at 540 nm using a VersaMax microplate reader (Molecular Devices) to assess airway haemorrhage. Lung lobes were either stored in RNAlater (Thermo Fisher Scientific), fixed in 10% neutral-buffered formalin for histology or minced and digested with Liberase (TL) low thermolysin concentration (Roche) for 30 min at 37°C for analysis of lung epithelial cells by flow cytometry and ex vivo stimulation of lung T cells using cell stimulation cocktail (plus protein transport inhibitors) (eBioscience). For ex vivo T cell IL-13 measurements, lung homogenates were cultured in the presence or absence of anti-CD3/CD28 for 72 h and culture supernatants were analysed for IL-13 by ELISA. On D6pi, adult small intestinal worms were counted using a dissecting microscope after incubation of the small intestine at 37°C to collect live adult worms that migrate out of the tissue.

### Flow cytometry

Single-cell suspensions were washed in PBS and Live/Dead staining (Thermo Fisher Scientific) was performed. Samples were Fc-blocked using $\alpha$-CD16/32 (2.4G2) (BD Biosciences) and mouse serum (Bio-Rad). Blocking and subsequent surface staining was performed using PBS containing 2 mM EDTA, 2% FBS, and 0.05% NaN$_3$. Antibodies used for staining are listed in Table 1. After surface staining, cells were incubated with IC fixation buffer (Thermo Fisher Scientific) before permeabilization for intracellular staining. For secondary detection of Ym1 and RELM-$\alpha$, Zenon goat and rabbit antibody labels (Thermo Fisher Scientific) were used. For RELM-$\alpha$ intracellular staining, cells were directly stained without stimulation or protein transport inhibition. Lung CD4$^+$ T cells were stimulated ex vivo with cell stimulation cocktail containing protein transport inhibitors (Thermo Fisher Scientific) for 4 h at 37°C before staining. For cell quantification, some samples were spiked with 10 μm polystyrene beads (Sigma-Aldrich) before acquisition. Data were acquired on a BD LSRFortessa flow cytometer and analysed using FlowJo v10 software.

### RNA extraction and quantitative real-time PCR

Tissue samples stored in RNAlater (Thermo Fisher Scientific) were processed for RNA extraction using a TissueLyser II and QIAzol reagent (QIAGEN). Isolated RNA was quantified using a Qubit fluorimeter and RNA BR kit (QIAGEN). cDNA was synthesized using

**Table 1. List of flow cytometry antibodies used.**

| Antigen | Clone | Manufacturer |
|---|---|---|
| CD45.2 | 104 | BioLegend |
| CD11b | M1/70 | BioLegend |
| CD11c | N418 | BioLegend |
| Ly6C | HK1.4 | BioLegend |
| Ly6G | 1A8 | BD Biosciences |
| Siglec-F | E50-2440 | BD Biosciences |
| TCR$\beta$ | H57-597 | BioLegend |
| CD3$\varepsilon$ | 17A2 | Thermo Fisher Scientific |
| CD4 | GK1.5 | BioLegend |
| CD8 | 53–6.7 | BioLegend |
| CD19 | 6D5 | BioLegend |
| B220 | RA3-6B2 | Thermo Fisher Scientific |
| EpCAM | 9C4 | BioLegend |
| CD31 | 390 | BioLegend |
| IL-4 | 11B11 | BioLegend |
| IL-5 | TRFK5 | BioLegend |
| IL-13 | eBio13A | Thermo Fisher Scientific |
| RELM-$\alpha$ | Polyclonal | Peprotech |

Tetro reverse transcription kit (Bioline) and oligo dT 15-mers (Integrated DNA Technologies). Quantitative real-time PCR was performed using SYBR™ green mix (Agilent Technologies) and a LightCycler 480 II (Roche). A list of primer sequences used are shown in Table 2. Gene expression levels were determined by second derivative maxima using standard curves (LightCycler software) and expressed relative to the housekeeping gene *Rpl13a*.

### Lung proteomic analysis

Samples from slides containing whole lung tissue sections were scraped excluding major blood vessels and processed as previously described (Herrera et al, 2020). Peptides were evaluated by liquid chromatography coupled tandem mass spectrometry using an UltiMate 3000 Rapid Separation LC system (Dionex Corporation) coupled to a Q Exactive HF mass spectrometer (Thermo Fisher Scientific). Raw spectra were aligned using MAXQuant software v1.6.17.0 (Cox & Mann, 2008) with the variable modifications of proline and methionine oxidation in addition to "matched between runs" being enabled. Raw data were then imported into R for differential analysis with MSqRob (Goeminne et al, 2018) using the default pipeline. Heat maps were plotted using scaled log$_{10}$-transformed LFQ counts.

### Hydroxyproline assay

Hydroxyproline assay was performed as previously described (Chang et al, 2020). Whole lungs were incubated overnight in 6 M HCl, in screw-top tubes at 100°C covered with aluminium foil. Tubes were cooled to RT and centrifuged at 12,000$g$ for 3 min.

**Table 2. List of primer sequences used.**

| Primer | Sequence (5′-3′) |
| --- | --- |
| *Ccl11* forward | CACGGTCACTTCCTTCACCT |
| *Ccl11* reverse | TGGGGATCTTCTTACTGGTCA |
| *Clca1* forward | CTGTCTTCCTCTTGATCCTCCA |
| *Clca1* reverse | CGTGGTCTATGGCGATGACG |
| *Foxa3* forward | GCTGACCCTGAGTGAAATCTAC |
| *Foxa3* reverse | ACGAAGCAGTCATTGAAGGAC |
| *Il4* forward | GAGAGATCATCGGCATTTTGA |
| *Il4* reverse | TCTGTGGTGTTCTTCGTTGC |
| *Il5* forward | ACATTGACCGCCAAAAAGAG |
| *Il5* reverse | CACCATGGAGCAGCTCAG |
| *Il13* forward | CCTCTGACCCTTAAGGAGCTTAT |
| *Il13* reverse | CGTTGCACAGGGGAGTCT |
| *Il33* forward | TCCAACTCCAAGATTTCCCCG |
| *Il33* reverse | CATGCAGTAGACATGGCAGAA |
| *Muc5ac* forward | GCATCAATCAACAGCGAAACTT |
| *Muc5ac* reverse | CGAGTCACCCCCTGAGTC |
| Nb-actin forward | GCATCCCGTGCTGCTGAC |
| Nb-actin reverse | GGCGTACAGCGACAACACTG |
| *Retnla* forward | TATGAACAGATGGGCCTCCT |
| *Retnla* reverse | GGCAGTTGCAAGTATCTCCAC |
| *Rpl13a* forward | CATGAGGTCGGGTGGAAGTA |
| *Rpl13a* reverse | GCCTGTTTCCGTAACCTCAA |

Hydroxyproline standards were prepared (starting at 0.25 mg/ml) and serially diluted with 6 M HCl. Samples and standards (50 μl) were transferred into Eppendorf tubes and 450 μl chloramine T reagent (55.79 mM chloramine T [initially dissolved in 50% N-propanol] in acetate citrate buffer—0.88 M sodium acetate trihydrate, 294 mM citric acid, 1.2% glacial acetic acid, and 0.85 M sodium hydroxide—adjusted to pH 6.5; reagents from Sigma-Aldrich) was added to each tube and incubated at RT for 25 min. Ehrlich's reagent (500 μl; 1 M 4-dimethylaminobenzaldehyde in N-propanol:perchloric acid [2:1]; Sigma-Aldrich) was added to each tube and incubated at 65°C for 10 min and absorbance at 558 nm was measured.

### Histological and immunofluorescence staining

Whole left lung lobes were paraffin embedded and 5 μm sections were prepared for haematoxylin/eosin or Masson's trichrome staining. For visualization of haemosiderin deposits, lung sections were rehydrated and stained with a solution of 5% potassium ferrocyanide with 10% HCl (Prussian blue) for 20 min; sections were then rinsed with dH$_2$O and counterstained with a solution of 1% neutral red and 1% acetic acid for 5 min before being rinsed with dH$_2$O and dehydrated for mounting. Bright-field images were captured using an Olympus slide scanner and analysed using

CaseViewer software (3DHISTECH). For immunofluorescence staining, lung sections were rehydrated and subjected to heat-mediated antigen retrieval using citrate buffer (10 mM sodium citrate, 0.05% Tween-20, pH 6.0) followed by primary antibody incubation overnight at 4°C using biotin-anti-Ym1 and anti-RELM-A (Table 1); sections were then incubated with anti-rabbit FITC (Invitrogen) and streptavidin NL-557 (R&D systems) for 30 min RT before being mounting using Fluoromount-G containing DAPI (SouthernBiotech). Fluorescent slides were imaged using an EVOS FL Imaging System (Thermo Fisher Scientific) and analysed using ImageJ. IL-33 was detected using a rabbit polyclonal anti-IL-33 antibody (ab118503; Abcam) and a donkey anti-rabbit IgG NorthernLights NL637-conjugated antibody (NL005; R&D systems). Integrated density (IntDen) per nuclei was calculated using ImageJ. First, eight-bit greyscale images were binarised using a minimum threshold of 15 to a maximum threshold of 255. This mask was then used with ImageJ's "Analyze Particles" function with a size limit of 10–100 pixels and a circularity limit of 0.25–1.00. Measurements of Area and IntDen were taken for each nuclei and the staining intensity was calculated as the IntDen—area for each nuclei. These intensity values were then averaged to give the mean nuclear staining across each regions of interest (ROI). For each animal, five ROIs were analysed and these were averaged to give the mean for that animal.

### Lung lacunarity analysis

Slide-scanned images of H&E–stained lung lobes were processed in a (KNIME) Konstanz Information Miner software workflow to obtain 50 random ROIs across the whole lung section. ROIs that contained lobe boundaries or extensive artefacts were excluded from the analysis. The ROIs were then converted to binary images and lacunarity was quantified using the FracLac plugin for ImageJ (default settings). Lacunarity values of all the ROIs were averaged to obtain estimates for the entire lobe.

### ELISA

BAL supernatants were analysed for RELM-α using commercially available ELISA kits (Peprotech). Analytes were detected using horse radish peroxidase-conjugated streptavidin and TMB substrate (BioLegend) and stopped with 1 M HCl. Final absorbance at 450 nm was measured using a VersaMax microplate reader (Molecular Devices).

### IL-4-Fc and IL-13-Fc injections

To extend the half-life of IL-4 and IL-13, fusion proteins were generated of mouse IL-4 and IL-13 with the Fc portion of IgG1 (custom order with Absolute Antibody). Mice were injected intraperitoneally with either PBS, 10 μg IL-4-Fc, or 10 μg IL-13-Fc in 100 μl PBS. In other experiments, mice were anaesthetised using isoflurane inhalation and intranasally instilled with either PBS, 10 μg IL-4-Fc, or 10 μg IL-13-Fc in 40 μl PBS. The following day at 18 h posttreatment, mice were culled for lung cell analysis by flow cytometry.

### Nanostring analysis

Quality control was performed on RNA samples with an Agilent 2200 TapeStation system before downstream analyses. Samples were diluted and 100 ng of RNA was processed for running on a Nanostring nCounter FLEX system using the Myeloid Innate Immunity v2 panel. Raw count data were imported into R for analysis using the limma package (Ritchie et al, 2015). Internal housekeeping and negative control probes were used to ensure data integrity and set thresholds for minimum expression. Data were normalised using the edgeR package (Robinson et al, 2009) and then differential expression was calculated using the limma package. Figures were generated in R using ggplot and the complexheatmap package. Heat maps were plotted using scaled normalised counts.

### Statistical analyses

Graphpad Prism 8 software was used for all statistical analyses. Data were assessed to be normally distributed by the D'Agostino-Pearson omnibus normality test. Differences between experimental groups were assessed by ANOVA (for normally distributed data) followed by Tukey–Kramer post hoc multiple comparisons test or an unpaired two-tailed $t$ test. In cases where data were not normally distributed, a Kruskal–Wallis test was used. For gene expression data, values were $\log_2$-transformed to achieve normal distribution. Comparisons with a $P$-value of <0.05 were considered to be statistically significant.

## Data Availability

The mass spectrometry proteomics data have been deposited to the ProteomeXchange Consortium via the PRIDE (Perez-Riverol et al, 2019) partner repository with the dataset identifier PXD021853.

## Supplementary Information

## Acknowledgements

This work was supported by the Wellcome Trust (203128/Z/16/Z, 110126/Z/15/Z, and 106898/A/15/Z) and the Medical Research Council UK (MR/K01207X/2). TE Sutherland was supported by Medical Research Foundation UK joint funding with Asthma UK (MRFAUK-2015-302). We thank Andrew McKenzie (Cambridge) for providing the $Il13^{tm3.1Anjm}$ mice. We further thank the Flow Cytometry, Bioimaging, Genomic Technologies, BioMS, and Biological Services core facilities at the University of Manchester.

### Author Contributions

AL Chenery: conceptualization, data curation, formal analysis, supervision, investigation, visualization, and writing—original draft, review, and editing.

S Rosini: conceptualization, data curation, formal analysis, investigation, visualization, and writing—review and editing.

JE Parkinson: data curation, formal analysis, investigation, methodology, and writing—review and editing.

J Ajendra: investigation, visualization, and writing—review and editing.

JA Herrera: methodology and writing—review and editing.

C Lawless: resources and formal analysis.

BHK Chan: investigation.

PA Loke: resources and writing—review and editing.

AS MacDonald: resources and writing—review and editing.

KE Kadler: conceptualization, supervision, funding acquisition, and writing—review and editing.

TE Sutherland: conceptualization, data curation, supervision, funding acquisition, and writing—review and editing.

JE Allen: conceptualization, data curation, supervision, funding acquisition, project administration, and writing—review and editing.

### Conflict of Interest Statement

The authors declare that they have no conflict of interest.

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
