## [Reviewer comments · Life Science Alliance]

Life Science Alliance

IL-13 deficiency exacerbates lung damage and impairs epithelial-derived type 2 molecules during nematode infection

Alistair Chenery, Silvia Rosini, James Parkinson, Jesuthas Ajendra, Jeremy Herrera, Craig Lawless, Brian Chan, P'ng Loke, Andrew MacDonald, Karl Kadler, Tara Sutherland, and Judith Allen

DOI: <https://doi.org/10.26508/lsa.202001000>

Corresponding author(s): Judith Allen, University of Manchester

Review Timeline:

Submission Date:	2020-12-19
Editorial Decision:	2021-01-03
Revision Received:	2021-04-15
Editorial Decision:	2021-05-20
Revision Received:	2021-05-29
Accepted:	2021-05-31

Scientific Editor: Shachi Bhatt

Transaction Report:

Please note that the manuscript was reviewed at Review Commons and these reports were taken into account in the decision-making process at Life Science Alliance.

Review
COMMONS

January 3, 2021

Re: Life Science Alliance manuscript #LSA-2020-01000-T

Prof. Judi Allen
University of Manchester
Unknown
Manchester
United Kingdom [GB]

Dear Dr. Allen,

Thank you for submitting your manuscript entitled "IL-13 deficiency exacerbates lung damage and impairs epithelial-derived type 2 molecules during nematode infection" to Life Science Alliance. We apologize for this delay in getting back to you, caused due to the fact that the LSA office was closed for holidays.

For a brief overview, this manuscript was reviewed via Review Commons, and the authors then decided to transfer the manuscript, the reviewers' comments and their revision plan to Life Science Alliance (LSA). The submitted material was reviewed by at least 2 editors at LSA and was deemed to be appropriate for further consideration provided the authors address the following, measurement of IL13 protein levels at D2 and D6 in infected WT mice, and the changes in IL23 and IL 33 + the data that the authors showed willingness to include in the revision in their rebuttal plan.

Thank you for this interesting contribution to Life Science Alliance. We are looking forward to receiving your revised manuscript.

Sincerely,

Shachi Bhatt, Ph.D.
Executive Editor
Life Science Alliance
<https://www.lsa-journal.org/>
Tweet @SciBhatt @LSAJournal

- A letter addressing the reviewers' comments point by point.
- An editable version of the final text (.DOC or .DOCX) is needed for copyediting (no PDFs).
- High-resolution figure, supplementary figure and video files uploaded as individual files: See our detailed guidelines for preparing your production-ready images, <https://www.life-science-alliance.org/authors>
- Summary blurb (enter in submission system): A short text summarizing in a single sentence the study (max. 200 characters including spaces). This text is used in conjunction with the titles of papers, hence should be informative and complementary to the title and running title. It should describe the context and significance of the findings for a general readership; it should be written in the present tense and refer to the work in the third person. Author names should not be mentioned.

B. MANUSCRIPT ORGANIZATION AND FORMATTING:

Dear Editors;

Please find below a point-by-point rebuttal to all the reviewer's comments. These include the original rebuttal updated to include the changes we have made to the manuscript, which are based on the additional experiments we performed in response to your requests. We believe we have addressed all the major points and the result is a much-improved manuscript.

Yours sincerely;

Judi Allen & Tara Sutherland, on behalf of all the authors.

Reviewer #1 (Evidence, reproducibility and clarity (Required)):

*This study addresses the role IL-13 in promoting lung damage following migration of the helminth *N. brasiliensis* larvae from the circulatory system to the lung. The work clearly shows using IL13^{-/-} mice that Nb elicited IL-13 immunity at days 2-6 post-infection reduces pathology. The authors demonstrate an association with reduced eosinophils but no effect on neutrophil numbers.*

Proteomic analysis identifies a number of molecules known to be involved in protecting against type 2 pathologies such as relm-a and SP-D.

The authors then identify a clear requirement for IL-13 in driving relm-a expression.

Finally, the authors present a whole lung RNA transcript profile which largely supports their proteomic observations.

Taken together the work presents a sound case for IL-13 being an important player in protecting against initial lung pathology.

****Major requests:****

The paper is really very interesting and important. To an extent it questions existing dogma of IL-13 being a driver of lung inflammation.

Addressing the following could hopefully be achieved using archived samples or with an acceptable amount of extra experimental work.

Figure 1: D2 and D6 Lung IL-13 concentrations (ELISA) in WT mice would set the scene for the papers story

We agree that showing IL-13 concentrations in the lung would nicely set the stage for the role of IL-13 during *Nippostrongylus* infection. In the original version, we showed IL-13 mRNA levels in Figure 3 but in this revised version, we include D2 and D6 mRNA data in Figure 1 (now Figure 1A, new text on manuscript line 117). We attempted to quantify IL-13 protein levels in the BAL fluid of infected WT mice on D2 and D6 post-infection. However, IL-13 in the BAL was below the levels of detection for our ELISA assay. Therefore, we performed new experiments to measure IL-13 protein in lung homogenates. However, IL-13 was also below the detection limit in whole lung homogenate but we were able to observe a progressive increase in T cell-derived IL-13 in the supernatant of anti-CD3/CD28 stimulated lung cell cultures (new Supplemental Figure 1, new text on line 119).

Figure 2: The authors should add evidence that function/activity of neutrophils/eosinophils is changed/not changed: e.g. granzyme, MBP, EPO release in BAL and/or lung.

As supported by referee 3, we feel that measuring functional readouts of neutrophils and eosinophils, while interesting, is currently outside of the scope of the paper. Further, with respect to eosinophils, we see a major reduction in total eosinophil numbers in IL-13-deficient mice which would likely result in a reduction in the level of functional molecules such as MBP. Thus, these readouts in the BAL may not be a reliable indicator of cellular function and results difficult to interpret in light of altered cell numbers.

Additionally, some data showing changes in epithelial stress related cytokines such as IL-23 and IL-33 would be informative (IHC and /or ELISA).

The reviewer makes a good suggestion that would complement our proteomics/pathway analysis. As described in our comment below regarding Foxa2 pathways, we do have additional data showing epithelial cell defects in the absence of IL-13 and this now added to a revised manuscript (New Fig 6 D,E). While we do see a trend for a reduction in IL-33 mRNA in infected IL-13-deficient mice, it is difficult to correlate this with functional protein. We therefore performed additional analyses to measure IL-23 and IL-33 protein levels by IHC of lung sections. We could not achieve adequate staining for IL-23 but were able to measure nuclear IL-33 in alveolar epithelial type II cells by immunofluorescence. Corroborating our mRNA data, we found that while infected WT mice had a significant increase in IL-33 staining in these cells on D2pi, nuclear IL-33 protein did not increase in the absence of IL-13 (new Figure 6E, new text on line 295). Notably, this is consistent with our finding that the Foxa2 pathway is up-regulated in the IL-13^{-/-} infected mice as Foxa2 regulates IL-33. This is now included in the manuscript discussion (Line 348)

The following will require a new experiment:

The authors present a strong case for RELM α being associated with/driven by IL-13 responses. The following I feel would prove that IL-13 driven RELM α is important in reducing lung pathology. Can enhanced lung pathology or cell responses associated with pathology be reduced/altered by dosing Nb infected IL13^{-/-} mice with recombinant relma or by restimulating BAL cells (for example) from IL-13^{-/-} mice. This team is well placed to comment on the potential for such an in vivo experiment to be feasible.

Or could the authors could also test the ability for other candidate molecules to reduce lung pathology? Would for example i/n dosing of IL-13^{-/-} mice with AMCase, BRP39 or SP-D protect against pathology? It would be expected to be the case for SP-D.

Our previous study has shown that RELM- α plays an important yet highly complex role during lung repair (see Sutherland et al. 2018: <https://doi.org/10.1371/journal.ppat.1007423>). The suggested experiments would advance our understanding of the function of RELM- α and other effector molecules during type 2 immunity and repair. However, it is unlikely that the impact of IL-13 will be due to a single effector molecule (as supported by Reviewer 3) and thus these types of experiments would shift the focus of the paper from the impact of IL-13 to understanding specific function of type 2 effectors. Since our study deals more broadly with the function of IL-13 rather than the downstream effectors, we hope that this will open up further investigation of these specific molecules to the wider community to take forward.

Reviewer #1 (Significance (Required)):

The manuscript places IL-13 as an important initiator of early protection from acute lung damage. This is important as it is to an extent a non-canonical role for this cytokine. This is also important as IL-13 can be manipulated therapeutically. To maximise potential application of such drugs requires detailed understanding of the various contextual roles of IL-13. This study provides such evidence.

The authors identify a range of target mediators.

This is an important body of work that is useful for understanding how acute lung damage can be regulated.

This work will be of interest to Type 2 immunologists, any researcher with an interest in pulmonary inflammation as well as mucosal immunity.

I make these suggestions/comments based on my own background in Type 2 immunity, lung inflammation and parasitic helminth infection and immunity.

Reviewer #2 (Evidence, reproducibility and clarity (Required)):

*In this manuscript, Chenery et al report that IL-13 plays a critical role in protecting mice from lung damage caused by the infection of a nematode, *Nippostrongylus Brasiliensis*, in WT or IL-13 knockout mice (IL-13 eGFP knock-in mice, Neill et al., Nature 2010). Phenotypically, they demonstrated that IL-13 genetic deficiency resulted in more severe lung injuries and haemorrhaging following the larvae migratory infections. Through the proteomic and transcriptomic profiling, they identified gene-expression changes involved in the cellular stress responses, e.g. up-regulating the expression of epithelial-derived type 2 molecules, controlled by IL-13. They also found that type 2 effector molecules including RELM-alpha and surfactant protein D were compromised in IL-13 knockout mice. Thus, they proposed that IL-13 has tissue-protective functions during lung injury and regulates epithelial cell responses during type 2 immunity in this acute setting. Overall, the manuscript was clearly written and a number of findings were interesting and expected compared to the published knowledge. However, this work could be improved and more impactful by further performing the following suggested experiments.*

Major points:

1. It may not be accurate to claim that "IL-13 played a critical role in limiting tissue injury ... in the lung following infection" since IL-13 participates in both repelling worms and activating tissue reparative responses. It is very hard to distinguish these two kinds of responses with the current experimental settings because the much higher worm burden led to more direct lung damage in IL-13^{-/-} mice than WT counterparts.

The reviewer raises an important point that we have now clarified in the revised manuscript. Based on several studies, the role of IL-13 in mediating *Nippostrongylus* expulsion occurs in the small intestine, after the parasites have already cleared the lung tissue. The number of worms in the lung do not differ at the time points we are investigating. We have qRT-PCR data measuring *Nippostrongylus*-specific actin levels, which we and others have previously shown to accurately reflect worm numbers. We can therefore demonstrate that the differences in lung damage do not reflect a difference in the number of larvae in the lungs of IL-13 KO mice compared to WT mice. These data have been incorporated into the manuscript to better clarify this point. Incorporated in Figure 1E, new text on line 140.

2. It would be more informative if the authors could perform the RNA-seq analysis on the IL-13-

responsive cell type such as airway epithelial cells (goblet cell) by comparing WT vs IL13-/- in the context of lung damage caused by Nitrostrongylus Brasiliensis infection.

RNA-sequencing of specific cells would indeed be an excellent experiment that would reveal more IL-13-dependent processes in our model. However, this would be a considerable undertaking at this stage (as reviewer 3 has pointed out). Nonetheless, our extended analysis of the Foxa2 pathway as requested below has highlighted a number of genes regulated by IL-13, which are known to be involved in epithelial cell function.

3. Figure 6C, the transcriptional profiling of mouse lungs revealed that the Foxa2 pathway was significantly up-regulated in the IL-13-/- infected mice. This is an important finding because this pathway plays a critical role in the process of alveolarization and inhibiting goblet cell hyperplasia. In order to validate this finding, some components in this pathway could be further examined.

We agree with the reviewer that showing additional validation data to support the Foxa2 defect in IL-13-deficient mice would strengthen our paper's overall message. We have now incorporated into the manuscript additional qRT-PCR data of IL-13-dependent genes regulated by Foxa2 (*Ctca1*, *Muc5ac*, *Ccl11*, and *Foxa3*) that clearly support this epithelial cell-specific defect. Shown in Figure 6D and new text on line 288.

Reviewer #2 (Significance (Required)):

Overall, the manuscript was clearly written and a number of findings were interesting and expected compared to the published knowledge.

****Referees cross-commenting****

To Reviewer #1's Review: fair and constructive

To Reviewer #3's Review: agree in general.

Reviewer #3 (Evidence, reproducibility and clarity (Required)):

In this study, Allen, Sutherland and colleagues utilize IL-13 deficient mice to investigate the function of IL-13 in the early response to lung tissue damage induced by helminth infection. They demonstrate that IL-13 deficiency has significant effects on the acute tissue response to helminth infection (at day 2 and 6 post-infection). Particularly, IL-13 deficiency results in increased lung hemorrhaging, and more pronounced lung tissue damage evidenced by increased gaps in the alveolar architecture. They perform proteomic and transcriptomic profiling of the lungs to determine IL-13-induced pathways and demonstrate many protein and gene expression changes in the absence of IL-13. These include dysregulated collagens, reduced epithelial-derived proteins RELM α and surfactant protein D, downregulated pathways related to cellular stress, and increased genes associated with the Foxa2 pathway.

Overall, the key conclusions are convincing, and the study design, methods and data analysis are clear, rigorous and thorough.

****Minor Comments:****

1. The authors concluded that lung epithelial cells are more sensitive to IL-13 than IL-4, but the intranasal injection of both proteins showed a similar induction of RELM α - investigation into this

difference would be useful. Alternatively, providing an explanation for these different findings could be helpful.

Our suggestion that epithelial cells are likely to be more sensitive to IL-13 was based both on our data and the existing literature. We would agree that we do not have direct evidence for this. Indeed, because the type 2 receptor can respond to both IL-4 and IL-13 this issue is difficult to easily resolve experimentally. Based on our new data in figure S4, we have expanded on this topic in the revised manuscript and provide two alternative explanations (line 264).

2. Providing data by immunofluorescence or flow cytometry of non-epithelial expression of RELM α following intranasal versus intraperitoneal injection of IL-4 versus IL-13 would be useful.

This is a good suggestion and we have additional flow cytometry data looking at hematopoietic cell expression of RELM- α from our experiments that we have now incorporated into the revised manuscript. We found that airway macrophages were another source of RELM- α in the lung and mirrored the airway epithelial cell responses to both intraperitoneal and intranasal delivery of IL-4 and IL-13. Shown in Figure S4, new text on line 254.

3. Discussion of IL-13Ra1 deficient mice would be useful, in particular the study by Karo-Atar and Munitz in Mucosal Immunology 2016, showing that IL13Ra1 is protective against bleomycin-induced pulmonary injury (PMID: 26153764). Comparing their data with the gene expression datasets from this study would be useful (acknowledging the caveat that IL-4 effects through the type 2 receptor would also be abrogated in these IL13Ra1 mice).

We agree that a comparison of IL-13Ra1 versus IL-13 deficiency should be included in the discussion of our manuscript and apologise for the omission. These authors found epithelial-specific defects in IL-13Ra1-deficient mice such as Clca1 (aka Clca3), RELM- α , and chitinase-like proteins even under homeostatic conditions, which is highly consistent with our data. This study also found that IL-13Ra1 deficiency led to increased bleomycin-induced pathology and together with our data, offers further insight into the IL-13/IL-13R α 1 axis during lung injury. Because of the limitations in trying to compare such distinct data sets we did not formally compare the gene expression data sets but have highlighted notable overlapping genes between the two mouse strains and disease models. We have added these points to our discussion. (Line 318).

4. The authors reference Chen et al. Nature Medicine 2012, but do not discuss the finding in this paper that neither IL-4 $^{-/-}$ nor IL13Ra1 $^{-/-}$ have increased lung hemorrhage. This might be a mouse strain issue and worthwhile discussing.

This is indeed a very important point, which we have now addressed in the revised discussion (Line 375). IL-4R α -deficient mice did show increased bleeding in the Chen et al. study that was not seen in the IL-13R α 1 KO suggesting IL-4 alone is sufficient to limit bleeding. This is in contrast to our study where we found increased bleeding in IL-13 KO mice independent of IL-4. However, a major difference between the studies is the background strain of mice used, which was BALB/c in the Chen et al. study versus C57BL/6 mice we used in our study. In addition to differences in IL-4 and IL-13 levels and responsiveness we have routinely observed that BALB/c are much more prone to bleeding, which is consistent with reports of enhanced pulmonary hemorrhage in BALB/c vs C57BL/6 mice (Fisher et al. 2016). Although we have yet to unravel these differences mechanistically, IL-4R α signals are important in vascular integrity (Knipper et al. 2015), which could explain differential requirements for IL-4 versus IL-13 to limit bleeding between the two strains.

5. Reference 32 and 36 (Sutherland PLoS pathogens) are duplicates

Our apologies, we have fixed the reference duplication.

Reviewer #3 (Significance (Required)):

This study addresses the specific function of IL-13 in acute helminth infection of the lung, which has not previously been studied, as most studies investigate the combined function of IL-4 and IL-13 through IL-4 receptor KO or Stat6 KO mice.

It is a thorough, well-conducted and well-organized study with high quality data using 'omics' strategies to profile IL-13-induced genes and proteins. Their data identifies intriguing pathways that are dependent on IL-13, opening new avenues to explore for IL-13-mediated protective roles in acute lung tissue damage. Therefore this study provides conceptual and technical advances. Additionally, since targeting IL-4 and IL-13 are in clinical trials or employed therapeutically for pulmonary disorders, the findings from these studies are clinically relevant. It would however have been useful to validate some of these pathways and demonstrate epithelial-specific outcomes for IL-13-induced tissue protection.

Previous studies using IL4RKO have shown that IL-4 and IL-13 are necessary to protect from acute tissue damage in helminth infection (Chen, Nature medicine - referred to by authors). Other studies have investigated IL-13 in fibrosis and granulomatous inflammation (papers referenced by authors, and Ramalingam Nature Immunology 2009). Last, one study shows that IL-13Ra1 signaling is important for protection in bleomycin-induced lung injury, findings using a different transgenic mouse, which are relevant for this study and may be useful to discuss (Karo-Atar, Mucosal Immunology 2016).

As stated above - the data in this manuscript identify intriguing pathways that are dependent on IL-13, opening new, exciting avenues to explore for IL-13-mediated protective roles in acute lung tissue damage. Their data is also unique as it combines proteomics and transcriptomics, and identifies previously unappreciated IL-13 regulated pathways such as cellular stress and Foxa2, which would be interesting to investigate further.

****Referees cross-commenting****

To Reviewer 1: The suggested data for Figure 1 (IL-13 concentrations) could be useful, but suggested experiments for Figure 2 could be outside the main focus of the paper.

For the main suggested experiment: treatment of IL-13^{-/-} with RELMalpha, this could be useful, One caveat is that RELMalpha might not be the only effector molecule downstream of IL-13 so the authors may not get a definitive answer. An alternative (although not as RELMalpha-specific) would be to treat IL13KO mice with FcIL-4 or FcIL-13 - the latter that drives RELMalpha, and look at whether FcIL-13 is more protective than FcIL-4.

We agree that rescue experiments could provide insights into the relative protective effects of IL-4 versus IL-13. However, it might be challenging to interpret the results in part because of the difficulty in establishing physiologically relevant doses and timing and the fact that IL-4 will also signal through the type 2 receptor. These difficulties are reflected in the interpretation of our current data as discussed above (pt. 1 reviewer 3). Although we have found IL-4 and IL-13 delivery

experiments valuable and have used them in many of our papers, we have always been cautious in our interpretation, as we typically use these at super-physiological doses.

To Reviewer 2: I agree with points 1 and 3 - especially with point 3, which would give more in-depth understanding into the functional outcomes of the IL-13 -> FoxA2 pathway identified. For point 2, RNA-seq of epithelial cells would be informative, but may be beyond the scope of the project.

May 20, 2021

RE: Life Science Alliance Manuscript #LSA-2020-01000-TR

Prof. Judith Allen
University of Manchester
Oxford Road
Manchester M139 9PT
United Kingdom

Dear Dr. Allen,

Thank you for submitting your revised manuscript entitled "IL-13 protects the lung and promotes epithelial-derived type 2 molecules during nematode infection". We would be happy to publish your paper in Life Science Alliance pending final revisions necessary to meet our formatting guidelines.

Along with the points listed below, please also attend to the following:

- please make sure the author order in your manuscript and our system match
- please add callouts for Figures S2A, B, and S4A-C to your main manuscript text
- please add scale bar to Figure S2A
- the manuscript title is different in our system vs on the manuscript file, please advise us on what is your preferred title for the paper

A. FINAL FILES:

-- Summary blurb (enter in submission system): A short text summarizing in a single sentence the study (max. 200 characters including spaces). This text is used in conjunction with the titles of

papers, hence should be informative and complementary to the title. It should describe the context and significance of the findings for a general readership; it should be written in the present tense and refer to the work in the third person. Author names should not be mentioned.

B. MANUSCRIPT ORGANIZATION AND FORMATTING:

Sincerely,

Shachi Bhatt, Ph.D.
Executive Editor
Life Science Alliance
<http://www.lsjournal.org>
Tweet @SciBhatt @LSAJournal

May 31, 2021

RE: Life Science Alliance Manuscript #LSA-2020-01000-TRR

Prof. Judith Allen
University of Manchester
Oxford Road
Manchester M139 9PT
United Kingdom

Dear Dr. Allen,

Thank you for submitting your Research Article entitled "IL-13 deficiency exacerbates lung damage and impairs epithelial-derived type 2 molecules during nematode infection". It is a pleasure to let you know that your manuscript is now accepted for publication in Life Science Alliance. Congratulations on this interesting work.

DISTRIBUTION OF MATERIALS:

Again, congratulations on a very nice paper. I hope you found the review process to be constructive and are pleased with how the manuscript was handled editorially. We look forward to future exciting submissions from your lab.

Sincerely,

Shachi Bhatt, Ph.D.

Executive Editor

Life Science Alliance

<http://www.lsjournal.org>
